# Neutrophil count multiplied by D-dimer combined with pneumonia may better predict short-term outcomes in patients with acute ischemic stroke

Yinting Xing[1,2]*, Wei Yang[1,2], Yingyu Jin[2], Yanhong Liu[1]*

1 Department of Clinical Laboratory, The Second Affiliated Hospital of Harbin Medical University, Harbin City, Heilongjiang Province, China, 2 Department of Clinical Laboratory, The First Affiliated Hospital of Harbin Medical University, Harbin City, Heilongjiang Province, China

* 15776687250@163.com (YX); liuusa2016@163.com (YL)

## Abstract

### Objective

To investigate the predictive value of neutrophil, D-dimer and diseases associated with stroke for short-term outcomes of acute ischemic stroke (AIS).

### Methods

By collecting the subitems of laboratory data especially routine blood and coagulation test in AIS patients, and recording their clinical status, the correlation, regression and predictive value of each subitem with the short-term outcomes of AIS were analyzed. The predict model was constructed.

### Results

The neutrophil count multiplied by D-dimer (NDM) had the best predictive value among the subitems, and the area under the receiver operating characteristic (ROC) curve reached 0.804. When clinical information was not considered, the Youden index of NDM was calculated to be 0.48, corresponding to an NDM value of 7.78, a diagnostic sensitivity of 0.79, specificity of 0.69, negative predictive value of 96%. NDM were divided into 5 quintiles, the five grade of NDM (quintile) were < = 1.82, 1.83–2.41, 2.42–3.27, 3.28–4.49, 4.95+, respectively. The multivariate regression analysis was conducted between NDM (quintile), Babinski+, pneumonia, cardiac disease and poor outcomes of AIS. Compared with the first grade of NDM (quintile), the second grade of NDM (quintile) was not significant, but the third grade of NDM (quintile) showed 7.061 times, the fourth grade of NDM (quintile) showed 11.776 times, the fifth grade of NDM (quintile) showed 23.394 times in short-term poor outcomes occurrence. Babinski sign + showed 1.512 times, pneumonia showed 2.995 times, cardiac disease showed 1.936 times in short-term poor outcomes occurrence compared with those negative patients.

**Data Availability Statement:** All relevant data are within the paper and its Supporting Information files.

**Funding:** The author(s) received no specific funding for this work.

**Competing interests:** The authors have declared that no competing interests exist.

## Conclusions

NDM combined with pneumonia may better predict short-term outcomes in patients with AIS. Early prevention, regular examination and timely intervention should be emphasized for patients, which may reduce the risk of short-term poor outcomes.

## Introduction

Acute ischemic stroke (AIS) is a disease with high morbidity and mortality, causes considerable disability-adjusted life-years worldwide [1, 2]. More seriously, numerous AIS patients have poor short-term outcomes after the onset of disease. For poor outcome events, early detection, diagnosis and treatment may greatly reduce the progression. At present, there are several related studies on laboratory projects as AIS prognostic indicators, but most of them are limited to one system, and little studies had combined coagulation, immune function and clinical data as predict models. Thus it is particularly important to seek predictors of such poor outcome events in new paradigm [3].

Laboratory test items, especially neutrophil and D-dimer, reflect changes which may lead to poor outcomes in AIS patients such as blood cells, coagulation status, immune biochemical stress and previous nutritional status [4–7]. The clinical manifestations and status of AIS patients would also affect their short-term outcomes. Therefore, in order to identify more practical predictors in the laboratory and provide tools for AIS clinical diagnosis and treatment, this study recorded the clinical manifestations, medical history and a series of laboratory items of AIS patients during hospitalization through follow-up to analysis, finally, a short-term predict model was established.

## Materials and methods

### Patient enrollment

Ischemic encephalopathy patients in the First Affiliated Hospital of Harbin Medical University during July 2019 to July 2021were enrolled in this retrospective research. AIS was diagnosed according to the Chinese Guidelines for the Diagnosis and Treatment of Acute Ischemic Stroke 2018. The patients were all elder than 18 years old (Fig 1). All patients were followed up for 1 month to chase the short-term poor outcome (defined as poor state, drowsiness, lethargy, coma, death, a minimum of just 1 of these criteria should be flagged as having a poor outcome) recorded in the medical history. Those whose clinical information or laboratory items data were incomplete would be excluded.

### Data collection of clinical information and laboratory items

**Clinical information collection.**   During hospitalization, basic information (sex, age) of AIS patients was collected. Since the incidence of other complications of stroke was low and difficult to be found in medical records, this study only selected common concomitant diseases and status for statistical analysis. It was focused on followings: transient ischemic attacks (TIA, a cerebral ischemia without lasting symptoms), diabetes, hypertension, cardiac disease, pneumonia, hyperlipidemia, hyperhomocysteine, aphasia, Babinski sign +, short-term poor outcomes recorded in medical history.

**Laboratory data collection.**   Laboratory data were collected from patients with ischemic encephalopathy tested within 2 hours of admission. In order to screen for predictive clinical laboratory items, common items in patients with ischemic encephalopathy were collected in

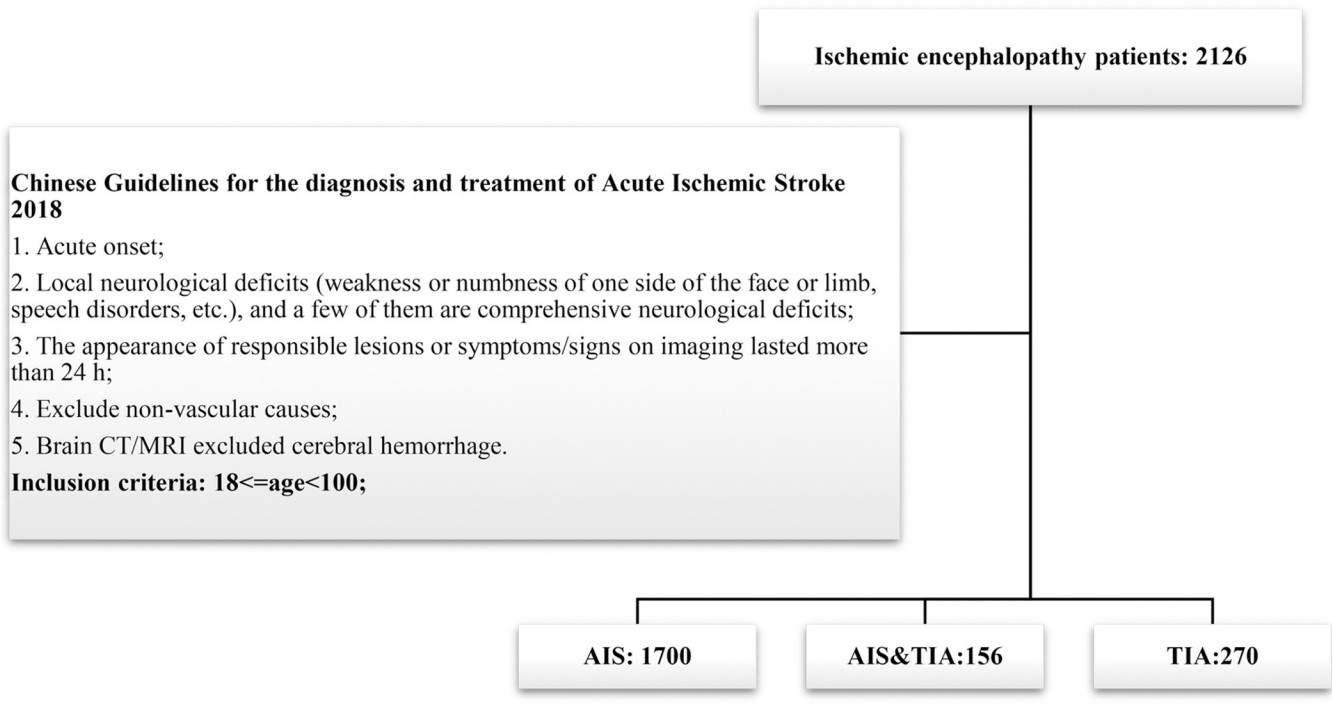

**Fig 1. The screening process of AIS patient enrollment.**

the preliminary stage of this study, including the routine blood (neutrophil count, NEU#, *10e +6/L; lymphocyte count, LYMPH#, *10e+6/L; monocyte count, MONO#, *10e+6/L; eosino-phil count, *10e+6/L; basophil count, BASO#, *10e+6/L; hemoglobin, HGB, g/L; red blood cell distribution width, RDW, %; platelet, PLT, *10e+6/L; platelet distribution width, PDW, %), coagulation and fibrinolysis (prothrombin time, PT, s; prothrombin activity, PTA, %; International Normalized Ratio, PTINR; activated partial thromboplastin time, APTT, s; fibrinogen, FIB, g/L; D-dimer, μg/mL(DDU)), chemical and immunology tests (homocysteine, HCY, μmol/L; albumin, ALB, g/L; proalbumin, PAB, mg/L; glucose, GLU, mmol/L; cholesterol, CHOL, mmol/L; triglyceride, TG, mmol/L; high density lipoprotein cholesterol, HDL-C, mmol/L; low density lipoprotein cholesterol, LDL-C, mmol/L; the ratio of LDL to HDL, LDL/ HDL ratio, %; apolipoprotein A, APOA, g/L; apolipoprotein B, APOB, g/L; the ratio of APOA to APOB, APOA/APOB ratio, %; lipoprotein a, Lpa, mg/L; folate acid, FOLATE, ng/mL; vita-min B12, B12, pg/mL). Among them, D-Dimer was the target item in this study. All detect method were shown in S1 Table.

**Sample size estimation.** Through previous experiments, it was found that the average value of D-dimer in well outcome patients' group is about 1.0, and the standard deviation (SD) was about 1.5. The average value of D dimer in poor outcome patients' group was about 3, and the SD was about 3. The detection unit was μmol/L (DDU), $\alpha = 0.05$ and $\beta = 0.2$, the test power was 0.8. Since the low incidence rate of short-term poor outcome in AIS, the ratio of sample size between poor outcome patients' group and well outcome patients' group was 1:10, and the proportion of missing samples was 0. The minimum sample size of two groups was calculated. The results showed that the poor outcome patients' group had at least 11 objects and the well outcome patients' group had at least 109 subjects, total in 120. The sample size of this study meets the requirements. In logistic regression analysis, according to the principle of 20 subjects corresponding to each item, 5 items would be included in the regression equation

in this study. Combined with the incidence rate short-term poor outcome of AIS, the total number of subjects in this study should be greater than 1800.

## Statistical analysis

According to whether the AIS patients had short-term poor outcomes, they were divided into Group P (poor outcomes) and Group W (well outcomes, indicates for no poor outcomes). The measurement data were expressed as the median (P25, P75), the counting data were described by N (%) The data were analyzed by a nonparametric test (Mann-Whitney U test) when comparing the distribution of baseline characteristics between the Group P and Group W. The measurement data were divided into 5 quintiles to obtain the grade data by SPSS visual binning, at the point of 20%, 40%, 60%, 80%. The correlation between subitems and poor outcomes of AIS was evaluated by Spearman's correlation coefficient for ranked data, and univariate and multivariate regression models were established. The measurement data was used as independent variable, the counting data was used as exposure variable. Then the ratio or product of significant subitems was calculated to check their relationship with the outcomes of AIS. Finally, the receiver operating characteristic (ROC) curve of subitems significantly related to poor outcomes were drawn to analyze which subitem was the most effective prognostic marker and to find a reasonable combination of clinical and laboratory subitems to predict the outcomes of AIS patients. A binary logistic regression forward LR test was used to screen multivariate items on the basis of a single factor $P<0.05$. The final multivariate regression equation was included to assess the items that affect the level of disability at the time of discharge. The true value of the ROC curve of poor outcomes was 0.5. The value corresponding to the maximum point of the Youden index (YI) was used to represent the optimal truncation, which represents the optimal balance between sensitivity and specificity.

The SPSS 19.0 (IBM Inc., USA) was used for statistical analysis. GraphPad Prism 7.0 Software (GraphPad Software Inc., USA) was used for graphics rendering. All of the tests were bilateral tests, $\alpha = 0.05$.

## Ethics statement

The studies involving human participants were reviewed and approved by the ethics committee in First Affiliated Hospital of Harbin Medical University (No. 202015). The patients provided their written informed consent to participate in this study.

## Results

### AIS patients in Group P accompanied with more clinical problems than in Group W

There were 1856 AIS patients in total (Fig 1), 120 (6.47%) patients had poor outcomes. Many of the patients had co-existing clinical problems such as TIA (8.4%), diabetes (27.7%), hypertension (56.8%), cardiac disease (20.6%), pneumonia (13.6%) and hyperlipidemia (19.9%) (Fig 2). Group P showed a higher percentage of patients suffering from co-existing clinical problems than Group W, such as cardiac disease (43.33% vs. 19.01%), pneumonia (45.00% vs. 11.41%), hyperhomocysteine (27.5% vs. 19.3%), aphasia (14.2% vs. 6.2%) and Babinski sign+ (53.3% vs. 36.6%), but interestingly, fewer patients in Group P suffered from TIA (0.83% vs. 8.99%) and hyperlipidemia (11.7% vs. 20.5%) than in Group W, $P<0.05$. The median age of the patients in Group P was 6 years elder than that in Group W (72 vs. 66, $P<0.05$), shown in Table 1. The median values of following laboratory items: NEU#, MONO#, RDW, PDW, PT, PTINR, FIB, D-dimer, HCY, GLU and Lpa in Group P were significantly higher than those in

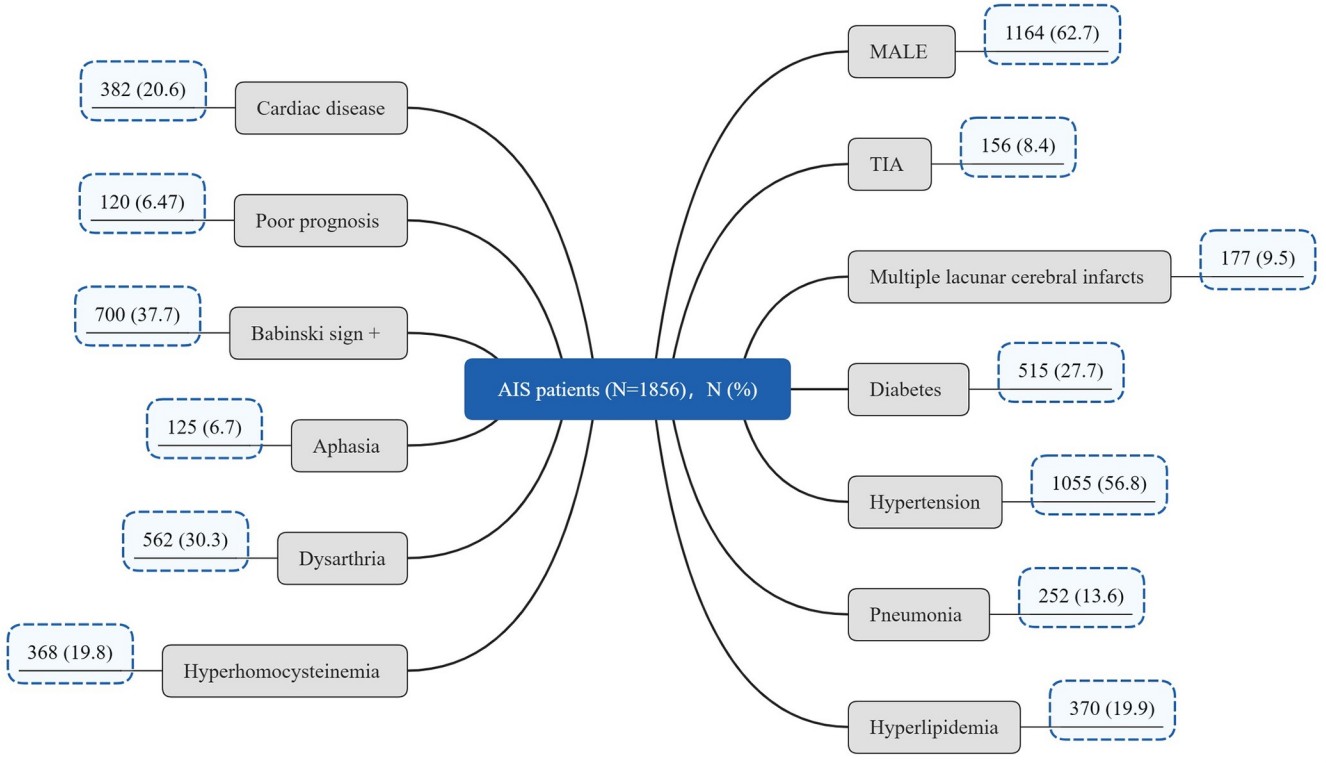

**Fig 2. The basic information of the AIS patients in the study.**

Group W. The median values of following laboratory items: LYMPH#, ESO#, PLT, PTA, ALB, PAB, TG, APOA and APOA/APOB in Group P were significantly lower than those in Group W, $P<0.05$. Shown in Fig 3, Table 1. Compared with W group, NEU# and D-Dimer in P group increased by 52.64% and 310.67%, respectively.

## The 5 quintiles result could be used to analyze the regression and draw the ROC curve

The measurement data were divided into 5 quintiles to gain the grade data. The differences of the grade data between Group P and Group W were checked. Among the grade data, only the difference of GLU and APOA were different from the comparison results of measurement data, the others were the same include NEU#, MONO#, RDW, PDW, PT, PTINR, FIB, D-DIMER, HCY, Lpa, LYMPH#, ESO#, PLT, PTA, ALB, PAB, TG and APOA/APOB ratio, $P<0.05$. The correlation test showed the same trend that they were correlated with the poor outcomes, regardless of the continuous data form or the 5 quintiles form, which indicated that the 5 quintiles result could be used to analyze the regression test and draw the ROC curve instead of the continuous measurement data. Shown in S2 Table.

## Multivariate regression tests showed that several subitems may predict the poor outcomes of AIS patients

Enter method was used for univariate regression analysis. Specific results were shown as Step A regression in S3 Table, in which only items with statistical significance were shown. Stepwise likelihood ratio method was used for multivariate regression analysis.

**Table 1. Compared result of significant subitems between Group W and Group P.**

| Subitems | Group W | Group P | CV%[#] | *P* value |
|---|---|---|---|---|
| NEU#, M (25, 75) | 4.54 (3.57, 5.96) | 6.93 (5.12, 9.07) | 52.64 | <0.001 |
| LYMPH#, M (25, 75) | 1.65 (1.25, 2.10) | 1.14 (0.81, 1.74) | -30.91 | <0.001 |
| ESO#, M (25, 75) | 0.09 (0.04, 017) | 0.05 (0.01, 0.13) | -44.44 | <0.001 |
| MONO#, M (25, 75) | 0.46 (0.36, 0.59) | 0.59 (0.39, 0.73) | 28.26 | <0.001 |
| RDW, M (25, 75) | 41.45 (38.50, 44.40) | 43.70 (39.83, 47.35) | 5.43 | <0.001 |
| PLT, M (25, 75) | 215.50 (177.00, 258.75) | 198.50 (151.25, 267.75) | -7.89 | 0.039 |
| PDW, M (25, 75) | 12.70 (11.20, 15.50) | 13.30 (11.80, 15.80) | 4.72 | 0.031 |
| PT, M (25, 75) | 11.90 (11.30, 12.60) | 12.55 (11.83, 13.40) | 5.46 | <0.001 |
| PTA, M (25, 75) | 94.00 (80.80, 105.60) | 77.60 (65.93, 92.40) | -17.45 | <0.001 |
| PTINR, M (25, 75) | 1.03 (0.97, 1.10) | 1.10 (1.04, 1.20) | 6.80 | <0.001 |
| FIB, M (25, 75) | 2.96 (2.41, 3.67) | 3.41 (2.37, 5.25) | 15.20 | 0.001 |
| D-dimer, M (25, 75) | 0.75 (0.50, 2.10) | 3.08 (1.27, 10.87) | 310.67 | <0.001 |
| Lpa, M (25, 75) | 49.75 (14.00, 175.87) | 113.00 (48.80, 412.80) | 127.14 | <0.001 |
| ALB, M (25, 75) | 37.88 (35.41, 40.16) | 36.70 (33.83, 39.59) | -3.12 | 0.027 |
| GLU, M (25, 75) | 5.69 (4.93, 7.40) | 6.14 (5.07, 7.77) | 7.91 | 0.046 |
| APOA, M (25, 75) | 1.13 (0.99, 1.28) | 1.07 (0.93, 1.24) | -5.31 | 0.019 |
| APOA/APOB ratio, M (25, 75) | 1.21 (0.98, 1.52) | 1.11 (0.87, 1.48) | -8.26 | 0.014 |
| PAB, M (25, 75) | 218.26 (186.80, 256.00) | 181.00 (144.00, 234.43) | -17.07 | <0.001 |
| TG, M (25, 75) | 1.41 (1.01, 1.97) | 1.22 (0.95, 1.67) | -13.48 | 0.007 |
| HCY, M (25, 75) | 13.15 (10.19, 18.10) | 16.28 (12.91, 25.00) | 23.80 | <0.001 |
| Age, M (25, 75) | 66.00 (58.00, 74.00) | 72.00 (64.25, 82.00) | 9.09 | <0.001 |
| Pneumonia, N (%) | 198 (11.41) | 54 (45.00) | 294.39 | <0.001 |
| Cardiac disease, N (%) | 330 (19.01) | 52 (43.33) | 127.93 | <0.001 |
| Aphasia, N (%) | 108 (6.2) | 17 (14.2) | 129.03 | <0.001 |
| TIA, N (%) | 156 (8.99) | 1 (0.83) | -90.77 | <0.02 |
| Babinski sign +, N (%) | 636 (36.6) | 64 (53.3) | 45.63 | <0.001 |
| Hyperlipidermia, N (%) | 356 (20.5) | 14 (11.7) | -42.93 | <0.019 |
| Hyperhomocysteine, N (%) | 335 (19.3) | 33 (27.5) | 42.49 | <0.029 |

Note: NEU#: neutrophil count, LYMPH#: lymphocyte count, ESO#: eosinophil count, MONO#: monocyte count, RDW: red blood cell distribution width, PLT: platelet, PDW: platelet distribution width, PT: prothrombin time, PTA: prothrombin activity, PTINR: International Normalized Ratio, FIB: fibrinogen, D-DIMER: D-dimer, Lpa: lipoprotein a, ALB: albumin, GLU: plasma glucose, APOA: apolipoprotein A, APOA/APOB ratio: The ratio of APOA to APOB, PAB: proalbumin, TG: triglyceride, HCY: homocysteine, TIA: transient ischemic attacks.

[#]CV% is the coefficient of variation; The calculation method was as follows: Quantitative data CV% = 100* (median of group P—median of group W)/median of group W, counting data CV% = 100* (percentage of group P—percentage of group W)/percentage of group W.

According to the results of univariate regression analysis, the first step multivariate regression analysis only included significant laboratory subitems. The results showed that NEU# (quintile), LYMPH# (quintile), MONO# (quintile), Age (quintile) and D-Dimer (quintile) could predict the poor outcomes of AIS patients. Specific results were shown in Step B regression in S3 Table.

The second step multivariate regression analysis, only clinical information was included. The results showed that TIA, and cardiac disease, pneumonia, Babinski sign+ had significant regression relationship with the outcomes in AIS. The results were shown in Step C regression in S3 Table.

The items included in the final step of multivariate regression analysis include the items with significant regression relationship in the first two steps of multivariate regression analysis.

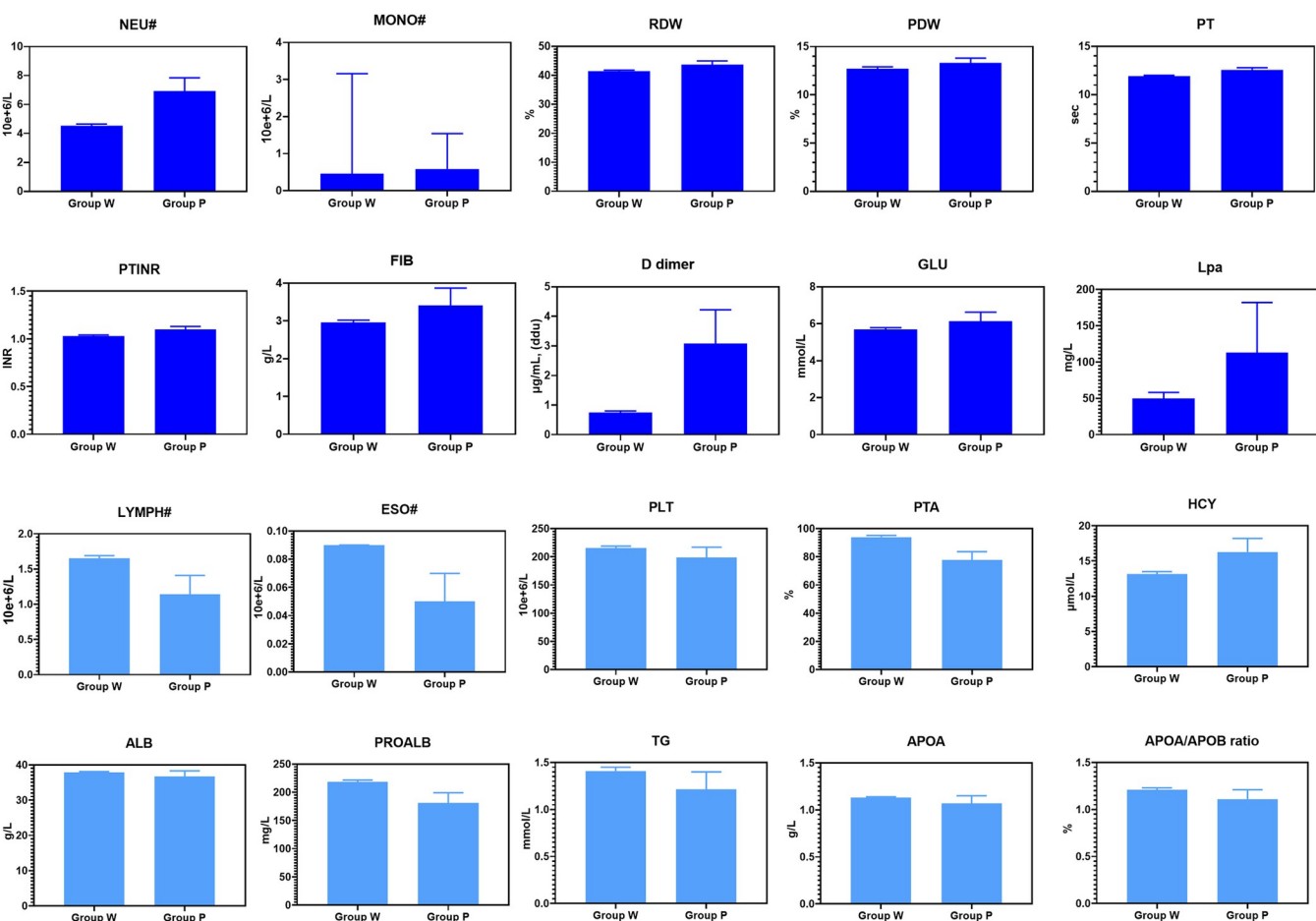

**Fig 3. Compared result of significant subitems between Group W and Group P (*P*<0.05*, median with 95 CI).** Items marked in dark blue are items higher in Group P than in Group W, and items marked in light blue are items lower in Group P than in Group W. This figure only shows the items with statistical significance.

The result showed that D-Dimer (quintile) (OR: 1.70, 95%CI: 1.40–2.07, P<0.05) could still independently predict poor short-term outcomes in AIS, and the incidence of poor short-term outcomes in AIS patients, which would increase by 70% for every quintile increase of D-dimer. The other subitems included NEU#(quintile) (OR:1.56, 95%CI:1.32–1.85, P<0.05), LYMPH#(quintile) (OR:0.77, 95%CI:(0.65–0.90, P<0.05), TIA (OR:0.17, 95%CI:0.02–1.25, P = 0.08), cardiac disease (OR:1.80, 95%CI:1.18–2.74, P = 0.01), pneumonia (OR:2.25, 95% CI:1.46–3.47, P<0.05), Babinski sign+ (OR:1.57, 95%CI:1.04–2.35, P = 0.03). These results suggested that AIS patients with increased D-Dimer (quintile), NEU#(quintile), combined with cardiac disease, pneumonia, and Babinski sign+ have an increased incidence of short-term poor outcomes, while AIS patients with increased lymphocyte count or combined with TIA have a decreased incidence of short-term poor outcomes. The results are shown in Step D regression in Table 2.

## Neutrophil count multiplied by D-dimer (NDM) had good regression relationship with poor outcomes of AIS patients

Among the subitems, NEU#(quintile), LYMPH#(quintile), D-DIMER (quintile), TIA, cardiac disease, pneumonia, Babinski sign+, the ROC curve showed that the most valuable subitem

**Table 2. Regression analysis of significant relative subitems to poor prognosis in AIS.**

| Subitems | *P value*[D] | OR[D] value (95% C.I. for adjusted OR[D]) |
|---|---|---|
| NEU# (quintile) | <0.001 | 1.56 (1.32–1.85) |
| LYMPH# (quintile) | <0.001 | 0.77 (0.65–0.90) |
| D-dimer (quintile) | <0.001 | 1.70 (1.40–2.07) |
| TIA | 0.08 | 0.17 (0.02–1.25) |
| Cardiac disease | 0.01 | 1.80 (1.18–2.74) |
| Pneumonia | <0.001 | 2.25 (1.46–3.47) |
| Babinski sign + | 0.03 | 1.57 (1.04–2.35) |

Note: [D]Forward likelihood ratio detection was performed after all significant results of steps B and C multivariate regression were included. NEU# (quintile): neutrophil count grade data, LYMPH# (quintile): lymphocyte count grade data, D-DIMER (quintile): D-dimer grade data by SPSS visual binning, at the point of 20%, 40%, 60% 80%. "OR": odds ratio. "C.I.": confidence interval.

was D-dimer, which covered an area under the curve (AUC) of 0.767, was no more than 0.8 (Fig 4A). Therefore, we considered accuracy could be improved by combining D-dimer, NEU# and LYMPH# into one score. The ROC curve was drawn again for the combined score. At the same time, logistic regression analysis was performed by combining the calculated values with statistically significant clinical information. Regression analysis showed that the neutrophil to lymphocyte ratio (NLR, P<0.001, OR: 1.137, 95%CI: 1.100–1.176), D-dimer to lymphocyte ratio (DLR, P<0.001, OR: 1.022, 95%CI: 1.012–1.033) and neutrophil count multiplied by D-dimer (NDM, P<0.001, OR: 1.002, 95%CI: 1.001–1.004) had a good regression relationship with poor outcomes of AIS patients. Multivariate logistic regression probability

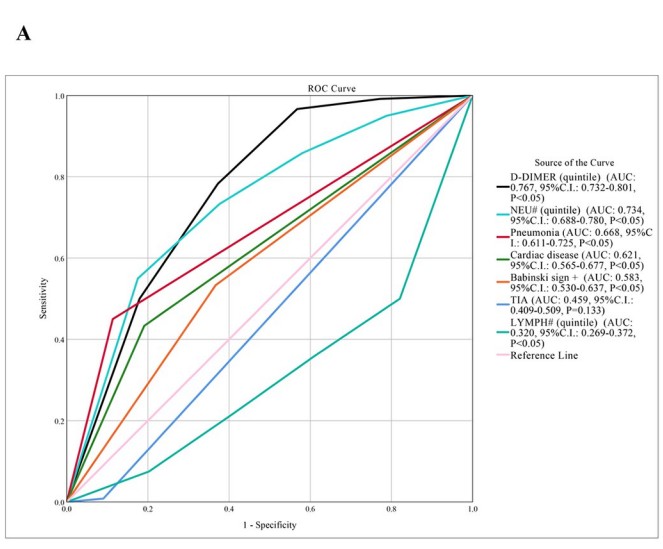

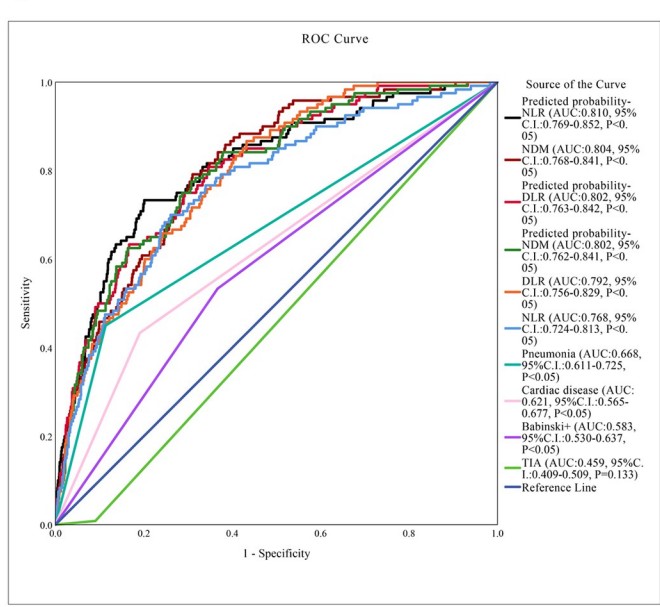

**Fig 4. The ROC curves.** A: ROC of TIA, Cardiac disease, Pneumonia, Babinski sign +, NEU# (quintile), LYMPH# (quintile), D-DIMER (quintile) to poor prognosis of AIS patients. NEU# (quintile): Neutrophil count in 5 quintiles, LYMPH# (quintile): Lymphocyte count in 5 quintiles, D-DIMER (quintile): D-dimer in 5 quintiles. B: ROC of NLR: The ratio of neutrophil count to lymphocyte count, DLR: The ratio of D-dimer to lymphocyte count, NDM: Neutrophil count multiplied by D-dimer, TIA: Transient Ischemic Attacks, Cardiac disease, Pneumonia, Babinski sign +, Predicted probability-NLR, Predicted probability-DLR, Predicted probability-NDM.

values were calculated for the three factors analyzed together with clinical information (cardiac disease, pneumonia, Babinski sign +, TIA), respectively. ROC curves were also drawn for the probability values. The final results were shown in Fig 4B. The NDM had the highest AUC without clinical information, while the NLR had the highest AUC when clinical information was included. When clinical information was not considered, the maximum approximate index of NDM was calculated to be 0.48, corresponding to an NDM value of 7.78, and the diagnostic sensitivity reached 0.79 at this point, while the specificity reached 0.69. Although the positive predictive value was only 25%, the negative predictive value could be as high as 96%.

## NDM (quintile) combined with pneumonia may better predict short-term outcomes in patients with acute ischemic stroke

NDM were divided into 5 quintiles, and multivariate regression analysis was conducted between NDM (quintile), Babinski sign+, pneumonia, cardiac disease and poor outcomes of AIS. The first grade of NDM (quintile) was < = 1.82, The second grade NDM (quintile)[1] was 1.83–2.41, The third grade NDM (quintile)[2] was 2.42–3.27, The fourth grade NDM (quintile)[3] was 3.28–4.49, The fifth grade NDM (quintile)[4] was 4.95+. The first grade of NDM (quintile) was the categorical covariable. The multivariate regression result showed a good regression relationship between NDM, Babinski sign+, pneumonia, cardiac disease and poor outcomes of AIS. Compared with the first grade of NDM (quintile), the second grade of NDM (quintile) was not significant, but the third grade of NDM (quintile) showed 7.061 times, the fourth grade of NDM (quintile) showed 11.776 times, the fifth grade of NDM (quintile) showed 23.394 times in short-term poor outcomes occurrence. Babinski sign + showed 1.512 times, pneumonia showed 2.995 times, cardiac disease showed 1.936 times in short-term poor outcomes occurrence compared with those negative patients. The predict model was shown in Table 3.

## Discussion

NDM, short for neutrophil count multiplied by D-dimer, was a new set of data relationships in this study. NDM combined with pneumonia may better predict short-term outcomes in patients with AIS. At the same time, combined with the clinical status, the advantages of test items as predictors of poor outcomes may be more fully utilized. The predict model in this study can be implemented as part of the delivery of care, to identify the patients who would be dangerous in the short-term period, and do the fit treatment to avoid the poor outcomes occurrence.

**Table 3. Regression analysis result of NDM (quintile) combined with pneumonia to the poor prognosis of AIS.**

| Items | Grade data of NDM | B | S.E. | *P value* | OR value (95% C.I. for adjusted OR) |
|---|---|---|---|---|---|
| NDM (quintile) | < = 1.82 | | | | |
| NDM (quintile)(1) | 1.83–2.41 | 0.378 | 0.917 | *0.680* | 1.460 (0.242–8.807) |
| NDM (quintile)(2) | 2.42–3.27 | 1.955 | 0.756 | *0.010* | 7.061 (1.604–31.077) |
| NDM (quintile)(3) | 3.28–4.49 | 2.466 | 0.738 | *0.001* | 11.776 (2.773–50.012) |
| NDM (quintile)(4) | 4.95+ | 3.152 | 0.729 | *0.000* | 23.394 (5.607–97.607) |
| Babinski+ | | 0.413 | 0.202 | *0.041* | 1.512 (1.017–2.247) |
| Pneumonia | | 1.097 | 0.213 | *<0.001* | 2.995 (1.973–4.544) |
| Cardiac disease | | 0.661 | 0.209 | *0.002* | 1.936 (1.285–2.916) |
| Constant | | -3.959 | 0.233 | *0.000* | |

Note: NDM (quintile): neutrophil count multiplied by D-dimer grade data, which was grouped into 5 quintiles. "OR": odds ratio. "C.I.": confidence interval.

Inflammation may induce secondary brain injury by aggravating BBB injury, microvascular failure, brain edema, oxidative stress and directly inducing neuronal cell death [8], and the detection of inflammation-related indicators as AIS severity and evaluation of prognosis has become a major direction of scientific research. Neutrophils may destroy host tissue, their deployment is strictly controlled by three main strategies: phagocytosis, degranulation, and release of Neutrophil extracellular traps (NETs). Neutrophils infiltrate into damaged brain tissue soon after the onset of AIS, leading to increased inflammation [9]. After ischemia/reperfusion, neutrophils accumulate in the meninges and perivascular spaces, eventually reaching the infarct parenchyma [10]. The researchers found NETs in AIS clots, which are thought to promote clotting and thrombosis. NETs in the ischemic brain parenchyma have been identified as the cause of secondary nerve injury [11]. D-dimer is the degradation product of fibrin during activation of fibrinolytic enzyme and is a marker of thrombin formation and fibrinolysis. D-dimer is relatively stable and the antibody is externally activated. In addition, the measurement of D-dimer is relatively simple, readily available and inexpensive. Therefore, among various biomarkers reflecting coagulation and fibrinolytic activation, D-dimer is the most commonly used test in clinical practice [12]. Early D-dimer levels are an independent predictor of large vessel occlusion and may contribute to better transfer of prehospital patients to appropriate stroke centers [13]. In this study, the two factors were combined by multiplication to amplify the predict effect on short-term poor outcomes of AIS. The AUC of the NDM was as high as 0.804, which was much higher than the NLR or D-dimer or neutrophil alone in this study. As the leader of independent predictors in this study, NDM has a sensitivity of 0.79 and a specificity of 0.69, and has a superior negative predictive value of 96%. There are many factors affecting neutrophil count and D-dimer, such as infection, injury, tumors and other problems [14–17]. When the NDM of clinical patients is greater than 7.78, it needs to be judged and analyzed in combination with other clinical states.

The laboratory test items are routinely performed after peripheral blood collection from AIS patients at admission. It has the advantages of minimal damage, fast detection and timeliness, and may objectively reflect the state of the body, which is significantly more convenient than imaging examination or clinical manifestations. The diagnostic value of an early outcomes may be realized by reflecting the various states of the body.

Some complications may arise from the brain injury itself, subsequent disability or immobilization, or AIS related treatments. These complications have a significant impact on the outcome of AIS patients and often hinder the recovery of the nervous system [18]. It is worth mentioning pneumonia, which is a prominent complication after AIS, accounting for 7–38%, as several clinical studies have confirmed that pneumonia may be independently associated with poor prognosis and disability in AIS patients [19–21]. This study also found that clinical conditions significantly related to the outcomes including TIA, cardiac disease, pneumonia and Babinski sign+. Previous studies have also shown that a variety of conditions are associated with the AIS severity and outcomes, such as myocardial infarction, stroke severity, prestroke disability, increased intracranial pressure, pneumonia and other complications [22, 23]. Through regression analysis in this study, the incidence of short-term poor outcomes in AIS patients with a history of TIA was 0.17 times higher than that in AIS patients without TIA, which may be related to their ability to be vigilant, review and treat blocked blood vessels frequently after having central nervous system problems. AIS patients with cardiac disease had 1.80 times higher and with pneumonia 2.25 times higher probability of poor outcomes. The reason for these phenomena may be that cardiac disease may reflect those serious problems in patients' blood vessels or microcirculation, so cerebrovascular itself is injured, and difficult to repair under stress. AIS triggers a multifaceted inflammatory response in the brain that contributes to secondary brain injury and infarct expansion [24]. Combined with pneumonia, AIS

patients have a more severe inflammatory immune response. AIS patients with a positive Babinski sign had 1.57 times higher probability of poor outcome, which also shows that the severity of the cerebrovascular disease affects the short-term outcomes. However, as predictors, their performance is lower than that of NDM.

When combining clinical and laboratory testing items, it was found that NLR combined had the best diagnostic value, with an AUC of 0.810, which was higher than the 0.802 of DLR and NDM, even though the latter two were not low. Studies have shown that short-term poor outcomes were more likely in female AIS patients, but this study found no significant difference in short-term adverse outcomes between sex [22, 25]. By dividing the laboratory test results into 5 quintiles, it was found that for each quintile of an increased NEU# result, the probability of poor outcomes would increase by 56%, for each quintile with increased LYMPH# results, the probability of poor outcomes would decrease by 23%, and for each quintile with increased D-dimer, the probability of poor outcomes would increase by 70%. This shows that inflammatory reactions and coagulation fibrinolysis may predict the occurrence of adverse outcomes. The probability of poor outcomes increased by 109% for each grade of NDM, which has not been previously reported. Many studies have utilized laboratory items alone as well as derived values, such as neutrophil-to-lymphocyte ratio (NLR) or platelet-to-lymphocyte ratio (PLR), but compare with those, the prediction model in this study showed more accurately by combined the clinical status and laboratory items together.

A previous study mentioned a problem with using D-dimer was the considerable variation in reporting mode, thus there is high potential for misreporting of D-dimer based on poor or incomplete reporting [26, 27]. In our study, we provided information including the unit (DDUs: μg/mL), reagent and machine, so that our study may be clearly comparable with other studies.

There are some limitations of this study. Follow-up studies should carry out multicenter studies, establish age groups, and conduct separate comparative analyses among different age groups. In addition, the data in this study are from high latitude areas in China, so research data from different geographies should be included in the subsequent research and then analyzed. Moreover, this study lack of information on other known predictors of poor AIS outcomes (e.g., stroke severity) since this information can be hardly found in the medical history.

## Conclusions

Neutrophil count multiplied by D-dimer (NDM) combined with pneumonia may better predict short-term outcomes in patients with AIS. The NDM could predict the short-term outcomes of AIS patients independently, with high sensitivity and specificity, and a high negative predictive value. The short-term outcomes of AIS were closely related to a variety of diseases and laboratory testing items. Patients' awareness of health management and regular physical examinations should be improved to reduce the incidence of short-term poor outcomes among AIS patients.

## Supporting information

**S1 Checklist.**
(DOCX)

**S1 Table. The detection materials and methods for laboratory subitems.**
(DOCX)

**S2 Table. Details of difference and correlated relationship analysis in quintile form of significant continuous measurement data between Group P with Group W.** This table only

shows the items that are significantly correlated with the poor prognosis of AIS. The items were grouped into 5 quintiles as grade variables (Q1-Q5). NEU#: Absolute neutrophil count, LYMPH#: Absolute lymphocyte count, MONO#: Absolute monocyte count, ESO#: Absolute eosinophil count, RDW: Red blood cell distribution width, PLT: Platelet, PDW: Platelet distribution width, PT: Prothrombin time, PTA: Prothrombin activity, PTINR: International Normalized Ratio, FIB: Fibrinogen, D-DIMER: D-dimer, HCY: Homocysteine, ALB: Albumin, PAB: Proalbumin, TG: Triglyceride, APOA/APOB ratio: The ratio of APOA to APOB, Lpa: Lipoprotein a. P value[1] indicates for the difference result by Mann-Whitney Test, P value[2] indicates for the correlated relationship by Spearman's rho Nonparametric Correlations. (DOCX)

**S3 Table. Regression analysis of significant relative subitems to poor prognosis in AIS in step A, B, C.** A: Enter method to check single item. B: Only significant laboratory items were included for forward likelihood ratio detection. C: Only clinically significant items were included for forward likelihood ratio test. "/" indicates that the factor is not included in the operation. NEU#: Absolute neutrophil count, LYMPH#: Absolute lymphocyte count, MONO#: Absolute monocyte count, ESO#: Absolute eosinophil count, RDW: Red blood cell distribution width, PLT: Platelet, PDW: Platelet distribution width, PT: Prothrombin time, PTA: Prothrombin activity, PTINR: International Normalized Ratio, FIB: Fibrinogen, D-DIMER: D-dimer, HCY: Homocysteine, ALB: Albumin, PAB: Proalbumin, TG: Triglyceride, APOA/APOB ratio: The ratio of APOA to APOB, Lpa: Lipoprotein a. "OR": Odds ratio. "C.I.": Confidence interval. (DOCX)

**S1 File.**
(SAV)

## Acknowledgments

Thank Mr. Zhiyuan Niu for his strong support in data collection, the laboratory department of the First Affiliated Hospital of Harbin Medical University for providing research sites and experimental equipment, and Ms. Meixi Bao of the ethics committee of the First Affiliated Hospital of Harbin Medical University for her guidance in ethics.

## Author Contributions

**Conceptualization:** Yinting Xing.

**Data curation:** Yinting Xing.

**Investigation:** Wei Yang.

**Methodology:** Yingyu Jin.

**Project administration:** Yanhong Liu.

**Software:** Yinting Xing, Yingyu Jin.

**Supervision:** Yanhong Liu.

**Writing – original draft:** Yinting Xing.

**Writing – review & editing:** Wei Yang.

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
