## [Decision Letter · Decision Letter 0]

22 Jun 2022

PONE-D-22-07312The product of neutrophils and D-dimer combined with post-stroke pneumonia may better predict short-term outcomes in patients with acute ischemic strokePLOS ONE

Dear Dr. Xing,

Thank you for submitting your manuscript to PLOS ONE. After careful consideration, we feel that it has merit but does not fully meet PLOS ONE’s publication criteria as it currently stands. Therefore, we invite you to submit a revised version of the manuscript that addresses the points raised during the review process.

Please pay special attention and emphasis to comments from both reviewers regarding choice of outcomes and their ascertainment. Furthermore, methodological details are warranted to ensure rigor and reproducibility. 

We look forward to receiving your revised manuscript.

Kind regards,

Farhaan S. Vahidy, Ph.D., M.B.B.S., M.P.H.

Academic Editor

PLOS ONE

Journal Requirements:

Reviewers' comments:

Reviewer's Responses to Questions

**Comments to the Author**

1. Is the manuscript technically sound, and do the data support the conclusions?

Reviewer #1: Partly

Reviewer #2: Partly

2. Has the statistical analysis been performed appropriately and rigorously? 

Reviewer #1: Yes

Reviewer #2: I Don't Know

3. Have the authors made all data underlying the findings in their manuscript fully available?

Reviewer #1: No

Reviewer #2: Yes

4. Is the manuscript presented in an intelligible fashion and written in standard English?

Reviewer #1: Yes

Reviewer #2: Yes

5. Review Comments to the Author

Reviewer #1: The authors report on a highly relevant topic. In general, it is recommended that the study design and methodology are described with more rigor such that the reader can feel confident in reproducing the study. Overall, the manuscript is written in reasonable fashion; however, there are minor grammatical errors and incomplete phrases that should be amended to improve readability. Additional review comments are provided in the attachment.

Reviewer #2: This paper highlights importance of D-Dimer + neutrophils (two culprits for post stroke poor outcomes)

1) Although definition of ischemic encephalopathy was clearly explained, transient ischemic attacks (TIA), multivariate lacunar cerebral infarcts (MLCI) were not defined

2) It is unclear why aphasia and Babinski sign were used as part of the statistical analysis compared to important physical exam findings (potentially some citations may be helpful)

3) The definition of poor short term outcomes defined as "poor state, drowsiness, lethargy, coma, death, in 1 month" is a bit vague as "poor state, drowsiness, lethargy" are not concrete markers and can vary with individual assessment

4) In terms of post stroke complications - confusion, depression, seizures, non chest infections are common culprits - Unclear why post stroke pneumonia was picked over the other common post stroke complications for analysis

5) Unclear why certain labs such as Vitamin B12/folate were collected and the significance to the main point of the paper

6. PLOS authors have the option to publish the peer review history of their article (what does this mean?). If published, this will include your full peer review and any attached files.

Reviewer #1: No

Reviewer #2: No

---

## [Author Response · Author response to Decision Letter 0]

20 Jul 2022

Response to reviewer 1:

Title: The product of neutrophils and D-dimer combined with post-stroke pneumonia may better predict short-term outcomes in patients with acute ischemic stroke

General comments:

In this cross-sectional analysis, Xing et al. compared the clinical and laboratory test profiles of acute ischemic stroke patients who experienced “well” versus “poor” short-term (1 month follow-up) outcomes, defined as poor state, drowsiness, lethargy, coma, or death. The authors report that among evaluated factors, the product of neutrophil and D-dimer levels as well as post-stroke pneumonia showed the strongest association with poor outcomes. The inclusion of a comprehensive panel of clinical conditions and laboratory tests are strengths of the study. In addition, the evaluation of multiple laboratory sub-items as well as derived ratios and products is systematic and provides a level of detail that is not included in all similarly focused studies. The authors also provide a thorough discussion of the potential biological underpinnings for their work and findings, which is advantageous for readers whose background is not in this domain.

In future iterations of this work, a more rigorous description of the study design and methodology is highly recommended, particularly in regards to how the data was abstracted for analysis and how the outcome and other variables were defined. In its current form, readers may have difficulty following along with the study’s design or potentially trying to reproduce the findings. Although the study of clinical and lab-based biomarkers is highly relevant to current research on predicting stroke outcomes, the value that the currently reported findings adds to the literature may be limited, due to its narrow study setting and design. The authors should clearly articulate the strengths of the current study’s findings in comparison to prior studies that have also evaluated the predictive utility of lab-based biomarkers (e.g., D-dimer levels alone, neutrophil-to-lymphocyte ratio [NLR], platelet-to-lymphocyte ratio [PLR]). There are minor grammatical / syntactical errors / incomplete phrases throughout the manuscript. It is suggested that the authors review to improve readability. 

Specific comments for each section of the manuscript are provided below.

Re: Thanks for the reviewer's recognition of this study and detailed suggestions. The author has corrected and replied one by one according to the suggestions

Introduction:

- The introduction can be kept brief and concise, but the authors should more clearly articulate what is known or has been reported about the utility of clinical or lab-based biomarkers for predicting stroke outcomes, as this is the stated focus of the study. This would also help the reader understand what research gaps that the current study is seeking to fill.

- Re: Content has been added in Introduction section.

Methods:

- Section 2.1: In addition to patient enrollment information, the authors should clearly describe the study design (e.g., prospective vs retrospective; cross-sectional) and define all outcomes and endpoints (e.g., primary, secondary), perhaps in a separate section of its own.

- Re: This was retrospective research; all outcomes and endpoints has been defined in the Section 2.1

- Section 2.2.1: This section lists out the clinical information collected during the hospitalization. At the end of the section, the authors briefly mention “short-term poor outcomes” and in parentheses list out the following: “poor state, drowsiness, lethargy, coma, death, in 1 month.” If these are the criteria for poor outcomes used in the study, it would be advantageous to provide more detail on how these data are collected and characterized (e.g., via discharge diagnosis, clinical notes). Is a minimum of just 1 of these criteria needed to be flagged as having a poor outcome? To improve readability, consider including these details in a separate section where you define the main study outcomes.

- Re: It has been changed to Section 2.1, and the way of collecting information has been clarified, and a minimum of just 1 of these criteria should be flagged as having a poor outcome.

- Section 2.2.2: Given the focus of the study on the predictive value of clinical and laboratory measures, the authors should clarify how the information was abstracted and transformed for analysis. Was the average of all values during the course of the hospitalization calculated? Include further details such that reproducibility of the study conduct is potentially easier.

- Re: All patients were followed up for 1 month to chase the short-term prognosis recorded in the medical history. Laboratory data were collected from patients with ischemic encephalopathy tested within 2 hours of admission.

- Section 2.2: Be very precise about the units measured for any clinical laboratory information. For example, in Section 2.2.3, reference values for D-dimer are given without units, and it is not clear if the detection units are the same.

- Re: The units had been added.

- Section 2.3: Related to comment above. The authors mention comparisons between “Group P (poor outcomes) and Group W (well outcomes)” but should clearly describe how these groups were defined and / or what threshold was used. Was categorizing poor vs well an either / or situation (i.e., any patient who did not have a “poor outcome” was considered a “well outcome”)?

- Re: In order to screen for predictive clinical laboratory items, common items in patients with ischemic encephalopathy were collected in the preliminary stage of this study. Group P (poor outcomes) and Group W (well outcomes, indicates for no poor outcomes) showed in section 2.3. 

- Section 2.3: For the regression analyses, was there a primary independent or exposure variable considered and defined?

- Re:The measurement data was used as independent variable, the counting data was used as exposure variable.

Results:

- Consider changing the heading for Section 3.2 or combine the findings with the previous section. It may be confusing to write that Section 3.2 is comparing the “performance of multiple laboratory items” when it is reporting descriptive differences.

- Re: The original 3.1 and 3.2 have been merged

- Section 3.3. The authors should describe the quintile analysis in more detail in the Methods. In Table 2, it is not clear which statistical difference that the P-value corresponds to. Is it referring to the difference in the continuous measurement data between Group P and Group W? Details for the quantitative results of the correlation test are not reported in the text. Overall, consider moving this table to the Supplement.

- Re: The continuous measurement data were divided into 5 quintiles to obtain the grade data by SPSS visual binning, at the point of 20%, 40%, 60% 80%. Table 2 had been transferred to supplemental table 2, the new P value1 indicates for the difference result by Mann-Whitney Test, P value2 indicates for the correlated relationship by Spearman's rho Nonparametric Correlations.

- Section 3.5. The authors make reference to “good regression relationship” between certain factors and poor AIS outcomes; however, the corresponding quantitative results that support this statement should be reported in the text. 

- Re: Regression analysis showed that the neutrophil to lymphocyte ratio (NLR, P<0.001, OR: 1.137, 95CI: 1.100-1.176), D-dimer to lymphocyte ratio (DLR, P<0.001, OR: 1.022, 95CI: 1.012-1.033) and neutrophil count multiplied by D-dimer (NDM, P<0.001, OR: 1.002, 95CI: 1.001-1.004), the message had been added in the section new section 3.4.

- Section 3.4. In general, the text for this section is very dense as it includes results for both univariate and multivariate analyses. Although it is understood that the results of the univariate analyses inform the stepwise approach taken for the multivariate models, it is suggested that the section be split up. Making this change would also improve the readability of Table 3 and eliminate much of the “white space” that is currently present for the 3 multivariate models on the righthand side.

- Re: The section had been split up into four parts. The original Table 3 was divided into to tables-table 2 and supplemental table 3. Then the “white space” had been eliminated.

- Section 3.5. In Figure 4, consider ordering the legend from highest to lowest AUC; otherwise, the lines which are drawn in close proximity as well as the overall quality of the image may make it difficult for the reader to distinguish the findings.

- Re: In Figure 4, the ordering the legend had been changed from highest to lowest AUC.

- Section 3.6. Provide the criteria for defining post-stroke pneumonia in the Methods. This analysis is also not well-described and should be made more clear.

- Re: Post-stroke pneumonia had been changed into pneumonia

- In general, make sure all tables and table footnotes include appropriate detail for the reader to understand. Look over the figures and ensure all axes and legends are complete.

- Re: All the tables and figures had been checked.

Discussion / Conclusion:

- The authors should include stronger discussion on how the study findings compare with current literature on the predictive value of lab-based biomarkers. For example, many studies have utilized D-dimer levels alone as well as derived values, such as neutrophil-to-lymphocyte ratio (NLR) or platelet-to-lymphocyte ratio (PLR). How does the neutrophil*D-dimer product (NDM) compare and can the authors provide an explanation for why it might serve as a better prognostic marker of poor outcomes?

- Re: The AUC of the NDM was as high as 0.804, which was much higher than the NLR or D-dimer or neutrophil alone in this study. As the leader of independent predictors in this study, NDM has a sensitivity of 0.79 and a specificity of 0.69, and has a superior negative predictive value of 96%.

- Consider additional limitations to mention. Although the focus of this study was on clinical and laboratory-based biomarkers, the authors should mention the lack of information on other known predictors of poor AIS outcomes (e.g., stroke severity).

- Re: This study lack of information on other known predictors of poor AIS outcomes (e.g., stroke severity) since this information can be hardly found in the medical history.

- The authors suggest that their findings may differ from others that are conducted in “different latitudes”. Does this refer to patient populations in different geographies? Or are there other nuances in study design that this refers to? Please clarify.

- Re: It refers to patient populations in different geographies

- Include a brief comment on how the study findings can be implemented as part of the delivery of care.

- Re: The predict model in this study can be implemented as part of the delivery of care, to identify the patients who would be dangerous in the short-term period, and do the fit treatment to avoid the poor outcomes occurrence. Many studies have utilized D-dimer levels alone as well as derived values, such as neutrophil-to-lymphocyte ratio (NLR) or platelet-to-lymphocyte ratio (PLR), but compare with current literature on the predictive value of lab-based biomarkers, the prediction model in this study showed more accurately by combined the clinical status and laboratory items together.

Response to reviewer 2:

Reviewer #2: This paper highlights importance of D-Dimer + neutrophils (two culprits for post stroke poor outcomes)

1) Although definition of ischemic encephalopathy was clearly explained, transient ischemic attacks (TIA), multivariate lacunar cerebral infarcts (MLCI) were not defined

Re: Transient ischemic attacks (TIA, defined as a cerebral ischemia without lasting symptoms); the status, multivariate lacunar cerebral infarcts (MLCI), was deleted as it was useless according to the statistical analysis results.

2) It is unclear why aphasia and Babinski sign were used as part of the statistical analysis compared to important physical exam findings (potentially some citations may be helpful)

Re: In this study, the incidence of other complications of stroke was low and difficult to be found in medical records. Therefore, this study only selected common concomitant diseases and states for statistical analysis, such as aphasia, Babinski sign +.

3) The definition of poor short-term outcomes defined as "poor state, drowsiness, lethargy, coma, death, in 1 month" is a bit vague as "poor state, drowsiness, lethargy" are not concrete markers and can vary with individual assessment

Re: The short-term prognosis was defined as poor state, drowsiness, lethargy, coma, death, a minimum of just 1 of these criteria should be flagged as having a poor outcome, that was recorded in the medical history.

4) In terms of post stroke complications - confusion, depression, seizures, non-chest infections are common culprits - Unclear why post stroke pneumonia was picked over the other common post stroke complications for analysis

Re: The incidence of other complications was low in this study and difficult to find in medical records

5) Unclear why certain labs such as Vitamin B12/folate were collected and the significance to the main point of the paper

Re: Laboratory data were collected from patients with ischemic encephalopathy tested within 2 hours of admission. In order to screen for predictive clinical laboratory items, common items in patients with ischemic encephalopathy were collected in the preliminary stage of this study, including Vitamin B12/folate.

---

## [Decision Letter · Decision Letter 1]

14 Sep 2022

Neutrophil count multiplied by D-dimer combined with pneumonia may better predict short-term outcomes in patients with acute ischemic stroke

PONE-D-22-07312R1

Dear Dr. Yinting Xing

We’re pleased to inform you that your manuscript has been judged scientifically suitable for publication and will be formally accepted for publication once it meets all outstanding technical requirements.

Kind regards,

Tomasz Urbanowicz

Academic Editor

PLOS ONE

Additional Editor Comments (optional):

Reviewers' comments:

Reviewer's Responses to Questions

**Comments to the Author**

1. If the authors have adequately addressed your comments raised in a previous round of review and you feel that this manuscript is now acceptable for publication, you may indicate that here to bypass the “Comments to the Author” section, enter your conflict of interest statement in the “Confidential to Editor” section, and submit your "Accept" recommendation.

Reviewer #1: All comments have been addressed

Reviewer #3: All comments have been addressed

2. Is the manuscript technically sound, and do the data support the conclusions?

Reviewer #1: Yes

Reviewer #3: Yes

3. Has the statistical analysis been performed appropriately and rigorously? 

Reviewer #1: Yes

Reviewer #3: Yes

4. Have the authors made all data underlying the findings in their manuscript fully available?

Reviewer #1: No

Reviewer #3: Yes

5. Is the manuscript presented in an intelligible fashion and written in standard English?

Reviewer #1: Yes

Reviewer #3: Yes

6. Review Comments to the Author

Reviewer #1: (No Response)

Reviewer #3: This is the revised version of a previously submitted study. The authors have correctly addressed all reviewers' questions and criticisms.

7. PLOS authors have the option to publish the peer review history of their article (what does this mean?). If published, this will include your full peer review and any attached files.

Reviewer #1: No

Reviewer #3: **Yes: **Mauro Silvestrini

---

## [Editor Report · Acceptance letter]

28 Sep 2022

PONE-D-22-07312R1 

Neutrophil count multiplied by D-dimer combined with pneumonia may better predict short-term outcomes in patients with acute ischemic stroke 

Dear Dr. Xing:

I'm pleased to inform you that your manuscript has been deemed suitable for publication in PLOS ONE. Congratulations! Your manuscript is now with our production department. 

Kind regards, 

on behalf of

MR Tomasz Urbanowicz 

Academic Editor

PLOS ONE